# Measurement Errors When Measuring Temperature in the Sun

**DOI:** 10.3390/s24051564

**Published:** 2024-02-28

**Authors:** Florian Teichmann, Alexander Pichlhöfer, Abdulah Sulejmanovski, Azra Korjenic

**Affiliations:** Institute of Material Technology, Building Physics and Building Ecology, Vienna University of Technology (TU Wien), 1040 Vienna, Austriaazra.korjenic@tuwien.ac.at (A.K.)

**Keywords:** radiation shield, radiation error, measurement error, air temperature measurement

## Abstract

In the validation of microclimate simulation software, the comparison of simulation results with on-site measurements is a common practice. To ensure reliable validation, it is crucial to utilize high-quality temperature sensors with a deviation smaller than the average absolute error of the simulation software. However, previous validation campaigns have identified significant absolute errors, particularly during periods of high solar radiation, possibly attributed to the use of non-ventilated radiation shields. This study addresses the issue by introducing a ventilated radiation shield created through 3D printing, aiming to enhance the accuracy of measurements on cloudless summer days with intense solar radiation. The investigation employs two pairs of sensors, each comprising one sensor with a ventilated and one with a non-ventilated radiation shield, placed on a south-oriented facade with two distinct albedos. Results from the measurement campaign indicate that the air temperature measured by the non-ventilated sensor is elevated by up to 2.8 °C at high albedo and up to 1.9 °C at a low albedo facade, compared to measurements with the ventilated radiation shield. An in-depth analysis of means, standard deviations, and 95% fractiles highlights the strong dependency of the non-ventilated sensor error on wind velocity. This research underscores the importance of employing ventilated radiation shields for accurate microclimate measurements, particularly in scenarios involving high solar radiation, contributing valuable insights for researchers and practitioners engaged in microclimate simulation validation processes.

## 1. Introduction

Temperature measurements play a pivotal role in assessing the impact of building greening on the local microclimate and in validating corresponding microclimate simulations. Recent investigations have highlighted substantial disparities between on-site temperature measurements and simulation outcomes, particularly when temperature sensors are exposed to direct solar radiation. This exposure results in an average absolute error ranging from 0.78 °C to 2.78 °C, with a standard deviation of absolute errors spanning 0.74 °C to 1.12 °C at 3:00 p.m. CEST [1,2,3]. However, the precise attribution of this absolute error to either the effects of direct solar irradiation or the inaccuracies within Computational Fluid Dynamics (CFD) simulation programs remains indistinct. Hence, a comprehensive examination of on-site temperature measurements is imperative to facilitate a realistic interpretation of CFD simulation results.

The intricate relationship between various environmental factors and temperature has been explored in numerous studies. Avraham et al. [4] posit a robust and rapid connection between air temperature measurements derived from radiance power and changes in absolute humidity. As emphasized by Erell et al. [5], the presence of short or long-wave radiation can introduce unacceptable errors in measurement accuracy. Direct sunlight on the temperature sensor’s surface heats the sensor, elevating the temperature around it compared to ambient air. To mitigate this, weather stations commonly employ radiation shields in front of temperature sensors. However, aging coatings diminish their ability to reflect sunlight, introducing errors up to 1.63 °C [6]. Moreover, reduced air circulation within non-ventilated radiation shields can impede the sensor’s response speed, resulting in a temperature error of 2–4 °C on clear, windless days at noon [7]. In summary, understanding the accuracy of temperature measurements necessitates the consideration of several environmental factors.

Research on radiation errors of fine-wire thermocouples (~0.1 mm diameter) within naturally aspirated radiation shields by Kurzeja [8] achieved a root-mean-square error of measured temperature at 0.35 °C and 0.16 °C, without and with correction, respectively. Nakamura and Mahrt [9], addressing the influence of wind speed and short-wave radiation on temperature measurements, proposed an empirical model with a root-mean-square error between measured and corrected values of 0.13 °C, albeit neglecting the impact of diffuse and direct solar radiation. Cheng et al. [10], analyzing temperature measurement errors between Chinese and imported radiation shields, achieved accuracies of 0.26 °C and 0.17 °C, respectively, with an improved correction method accounting for global solar radiation and wind speed effects on air temperature. Liu et al. [11] proposed a novel natural ventilated radiation shield, employing a back-propagation neural network algorithm to reduce radiation errors to within ±0.1 °C at wind speeds exceeding 1.5 m/s. These results are comparable to those of ventilated radiation shields, which typically achieve a root-mean-square error of less than 0.2 °C [12,13].

Despite advancements in traditional radiation shields, a significant error range persists in measurements with direct and indirect solar radiation, primarily dependent on wind speed. This study aims to investigate the influence of wind speed on the accuracy of non-ventilated sensors. Two pairs of sensors, one equipped with a non-ventilated radiation shield and the other with a ventilated radiation shield, were mounted on a south-oriented facade with varying albedos. Simultaneously, wind speed measurements were conducted atop the test stand. The subsequent section details the test equipment and research setting.

## 2. Materials and Methods

The ventilated radiation shields used in this study were developed at the TU Wien in 2022 to reduce measurement errors in temperature measurements with the influence of radiation. To produce these radiation shields, a total of 7 plastic parts were printed using a 3D printer (XYZ resolution 12.5, 12.5, 5 microns). These can be assembled together with an electric fan (60 × 60 × 15 mm, 4200 rpm, 0.58 m^3^/min) and white-painted aluminum housing (sheet thickness 2 mm, diameters 60 mm and 80 mm) to form a radiation shield (see Figure 1). The aluminum housing provides effective thermal insulation and is less likely to heat up than non-ventilated radiation shields. The thermal insulation in ventilated radiation shields is achieved by airflow between two aluminum tubes. This helps reduce the heat generated by external solar radiation and allows the sensor inside the tube to accurately measure the ambient temperature. The size and power of the fan was chosen to provide a practical size for the radiation shield and to ensure an airflow speed of at least 3 m/s at the sensor.

The measurements were carried out at the outdoor test stand at the Campus Science Center of the Vienna University of Technology (see Figure 2). The test stand is a free-standing test building at 192 m above sea level (Adriatic Sea). The nearest building to the south is 10 m high and 40 m away. The south-facing facade of the test stand was painted half white and half black. Four temperature sensors were mounted on both sides: one sensor in a conventional non-ventilated radiation shield, one sensor in a ventilated radiation shield, one free-hanging sensor, each at a distance of 30 cm from the facade, and one surface temperature sensor (see Figure 3 and Figure 4). The reference air temperature was measured in the 55 cm gap below the test stand in the shade. The wind velocity and direction were measured by a weather station on top of the test stand.

The following instruments were used:Data logger, exporting the measurement data once a week;Temperature and humidity probes for temperatures from −40 to 60 °C and an accuracy at 23 °C of ±0.5%rh and ±0.1 °C used for the sensors T1 to T4 as well as the reference temperature sensor;PT1000 probes (accuracy ±0.3 °C) used for the air temperature sensors without radiation shields as well as the surface temperature sensors;Weather station for wind measurement, using a solid-state magnetic sensor for wind speed (resolution 0.4 m/s and accuracy ±0.9 m/s or ±5%) and a wind vane with potentiometer for wind direction (display resolution 22.5° and accuracy ±3°).

The temperature and humidity probes were calibrated at a temperature of 30 °C and a humidity of 35, 50, and 70%rh using a humidity generator with a stability of 0.1%rh and 0.01 °C. The measurements at the test stand were carried out in summer 2023 from 8 July to 27 August. After the measurements, all sensors were placed inside the test stand from 6 September to 11 September. Inside the test stand, the temperature was set to 22 °C. All sensors recorded the same temperature.

Following the main measurements, another simplified measurement setup was installed to evaluate the validity of the reference temperature measurements under the test stand. For this purpose, three sensors were mounted next to each other centrally under the test stand, two of them without a radiation shield and one of them with a non-ventilated radiation shield. The temperatures were measured from 18 September to 20 September and compared to the temperature measurements of the ventilated sensor on the black facade (sensor Black-T1-v).

The evaluation of the measurement data is provided in the following section.

## 3. Results

The summer of 2023 was the seventh warmest summer in measurement history of Austria, with an average temperature 1.1 degrees above the average of the recent past (1991–2020) and 2.8 degrees above the mean compared with the 1961–1990 climate period [15]. The course of the air temperature (sensor Ref-AT) as well as the solar radiation (measured at a weather station in Unterlaa, Vienna, in a distance of 5 km) are shown in Figure 5. A comparison of the mean values of the air temperatures in front of the facade of the test stand for the entire measurement period with the mean temperature of the reference sensor under the test stand shows clear deviations for the freely suspended sensors without radiation shield (White-AT and Black-AT) of about +1.4 degrees (see Figure 6). For the sensors with radiation shielding (ventilated and non-ventilated), the mean deviations were within 0.1 degrees, with the sensor White-T3-v being the only sensor with a mean temperature below the temperature of Ref-AT.

For a detailed analysis of the temperature deviations of the different sensors in front of the test facade, in this study two representative model days were selected, which should represent summer days with clear skies, peak outdoor air temperatures, and high solar irradiation. The selected days were 27 July and 21 August. On 27 July, a moderate summer day with high solar irradiation, the reference temperature Ref-AT reached a maximum value of 26.2 °C and the global radiation was up to 910 W/m^2^. On the other hand, 21 August was a hot summer day with a peak air temperature of 36.5 °C and a maximum global radiation of 779 W/m^2^.

When looking at the temperature curves of the first reference day on 27 July (see Figure 7), a maximum temperature of the sensors White-AT and Black-AT (without radiation shield) of up to 32.5 and 32.8 degrees, respectively, can be seen. With a ventilated radiation shield, temperatures of only up to 27.4 degrees were measured in front of the black facade (Figure 7a), whereby the difference from the non-ventilated radiation shield was −0.3 degrees. In front of the white facade (Figure 7b), the maximum temperature with ventilated and non-ventilated radiation shields was 27.1 and 28.1 degrees, respectively. Compared to the maximum reference temperature under the test stand of 26.2 °C, all sensors in front of the facade showed higher maximum values. A contrasting picture emerges during dusk, night, and dawn, from approximately 6 p.m. to 10 a.m. During these hours, the facade sensors measured temperatures up to 0.4 degrees lower than the reference sensor under the test stand. This could be due to the reduced sky view factor of the reference sensor and the long-wave radiation exchange between the test stand floor and the ground.

The second reference day, 21 August, showed a similar course of temperatures, but with higher absolute values (see Figure 8), with a maximum reference temperature under the test stand reaching up to 36.5 °C. Also, the air temperatures in front of the white and black facade were elevated in both cases to 42.2 and 42.3 °C, respectively, without a radiation shield (White-AT and Black-AT); to 38.4 and 37.8 °C, respectively, with a non-ventilated radiation shield; and to 36.4 and 36.6 °C, respectively, with a ventilated radiation shield.

Peak values of up to 36.7 and 47.7 degrees were measured on the surface of the white facade on 27 July and 21 August, respectively. On the surface of the black facade, temperatures reached up to 58.3 and 70.1 degrees on the two reference days, which was 21.6 and 22.4 degrees above the surface temperature values of the white facade.

The absolute deviation in the measured air temperatures of the non-ventilated sensors from the ventilated sensors in front of the black and white facade (Black-T2-nv–Black-T1-v and White-T4-nv–White-T3-v, respectively) is shown in Figure 9. On both reference days, higher daytime temperatures were recorded by the sensors with the non-ventilated radiation shield than by the sensors with the ventilated radiation shield. The deviation of up to 2.1 and 2.3 degrees in front of the white facade on 27 July and 21 August, respectively, was always greater than that for the sensors in front of the black facade, where the temperature differences were peaking at 1.0 and 1.6 degrees, respectively. The mean daytime temperature differences from 10 am until 6 pm were 0.2 and 0.8 degrees for the black and 0.7 and 1.2 degrees for the white facade on 27 July and 21 August, respectively. The higher values of temperature differences between ventilated and non-ventilated sensors in front of the white facade can be attributed to the higher reflection of solar radiation, leading to an increase in the radiation error of the non-ventilated radiation shields.

During the night hours, however, the non-ventilated sensors tended to record slightly lower temperatures, which were, on both reference days, up to 0.3 and 0.4 degrees below the temperatures of the ventilated sensors in front of the white and black facade, respectively. This could be due to the effect of long wave radiation exchange with the sky, which becomes negligible in the case of ventilated radiation shields.

Compared with the reference temperature Ref-AT below the test stand, large deviations can be seen for both the non-ventilated and the ventilated sensors on both reference days (see Figure 10). The maximum deviation was 2.9 and 3.2 degrees between 2 p.m. and 4 p.m. local time on 27 July and 4.1 and 4.8 degrees on 21 August for the non-ventilated sensors in front of the black and white facade, respectively. For the ventilated sensors, the maximum deviation was still 1.9 and 2.0 degrees, respectively, on 27 July and 2.9 and 2.6 degrees, respectively, on 21 August.

On the other hand, during the night hours, between 7 p.m. and 9 a.m., the measured temperatures on the facade were up to 0.4 and 1.9 degrees lower than under the test stand on 27 July and 21 August, respectively.

An analysis of the correlation of the temperature readings of the non-ventilated and the ventilated sensors over the entire measurement period shows for the white as well as the black facade a decrease in the correlation with increasing air temperature, with the scatter of the measured data increasing at the same time (see Figure 11). The reason for this is expected to be the increase in radiation error at increasing temperatures, which is more significant for the non-ventilated sensors than for the ventilated sensors. Comparing the white facade with the black facade, the radiation error is more significant in front of the white facade, with the trend line for the temperature difference between the ventilated and the non-ventilated sensor reaching 0.62 °C at an air temperature (Ref-AT) of 36 °C. At the same air temperature, in front of the black facade the temperature difference was just 0.16 °C (see Figure 12). The big scatter in the temperature differences at higher temperatures might also be explained by the increasing impact of the wind speed on the radiation errors: at low temperatures and subsequently low solar radiation, the radiation error is small, regardless of the wind speed; at higher temperatures, the radiation error remains low at high wind speeds—at low wind speeds, though, the radiation error increased proportional to solar irradiation. In all graphs in Figure 11 and Figure 12, a red dotted line shows the linear trend line of the data, with the formula and the coefficient of determination shown in the bottom right.

To assess the significance of the correlation between Ref-AT and the difference between “Black-T2-nv” and “Black-T1-v” as well as “White-T4-nv” and “White-T3-v”, Student’s *t*-test was used together with the Pearson correlation. The Pearson correlation showed a weak positive correlation between Ref-AT and the temperature differences for the white facade (r = 0.1975) and a medium correlation for the black facade (r = 0.4683). Student’s *t*-tests for dependent samples, between the measured values Black-T1-v and Black-T2-nv as well as between the measured values White-T3-v and White-T4-nv, showed a significant difference between the mean values. For Black-T1-v and Black-T2-nv the t-statistic was −13.68, while for White-T3-v and White-T4-nv the t-statistic was −46.39. In both cases, the t-statistics were significantly larger than the critical t-values for both one-sided and two-sided tests. This indicates that the difference between the means was significant in both cases. The *p*-values were extremely low (very close to or equal to zero), indicating that the null hypothesis that there is no difference between the means is rejected. Overall, the results of both t-tests show that there is a statistically significant difference between the mean values of the measurements with ventilated and non-ventilated radiation shields.

The correlation of the measured temperatures in front of the facade and under the test stand over the entire measurement period is shown in Figure 13 for the ventilated and non-ventilated sensors with linear trend lines for both the black and the white facade data. Again, the correlation was decreasing and the scatter increasing at higher temperatures, with the scatter of the non-ventilated sensors being greater.

In each case, the results of the Student’s *t*-test yielded t-statistics well outside the critical range for both one-sided and two-sided tests, and *p*-values close to zero, indicating that the probability of obtaining such results purely due to chance is negligible. Therefore, we can reject the null hypothesis and conclude that statistically significant differences exist between the mean values of the reference temperature Ref-AT and the temperatures Black-T1-v, Black-T2-nv, White-T3-v, and White-T4-nv.

The means, standard deviations, and 95% fractiles of the relative temperature differences of the ventilated and non-ventilated sensors in front of the white and black facade, as well as the respective differences to the reference sensor Ref-AT for the entire measurement period, are listed in Table 1. The mean values show very small temperature differences between −0.04 and +0.13 degrees due to the described phenomenon that the differences change the plus/minus sign at night hours. Accordingly, the standard deviations showed higher values between 0.28 and 1.01 degrees. Additionally, the 95% fractiles are included, stating that there is a 95% probability that the actual temperature differences will not exceed these values. In total, it can be stated that, in the case of non-ventilated radiation shields, the temperature differences, and thus the radiation errors, are bigger in front of the white facade compared to the black facade. As stated above, this phenomenon can be attributed to the additional reflections of solar radiation reflected from the wall onto the sensors. Due to the higher reflectivity of the white facade, the influence on the radiation error is bigger. On the other hand, in the case of ventilated radiation shields, the radiation error was slightly bigger in front of the black facade, as the reflections from the facade became negligible due to the active ventilation of the radiation shields. Instead, the effect of the black facade heating up more due to a higher absorption of solar radiation and heating up the air in front of the facade at the same time seems to be predominant in this case.

Based on the assumption that large scattering at higher temperatures, as seen especially in Figure 12, might be related to the current wind strength, the wind strength was also recorded from 13 August. The wind speed was recorded every ten minutes with one digit after the decimal point. Its course is shown in Figure 14, together with the reference air temperature Ref-AT. The highest wind speed recorded was 5.4 m/s, and the mean wind speed was 1.27 m/s, with the mean wind direction being approximately south (177°). Peak wind speeds were generally during daytime, frequently coinciding with peak temperatures.

With regard to the correlation of the wind speeds with the difference between the temperatures in front of the facade and underneath the test stand, there was a clear tendency for temperature differences to be lower at higher wind speeds (see Figure 15). This was most obvious for the non-ventilated sensor in front of the white facade, which showed the largest radiative errors at low wind speeds. At wind speeds close to zero, the temperature differences were often negative, which was due to the phenomenon observed during night hours with low wind, where temperatures below the test stand were higher than in front of the facade. Therefore, these measurement data should not be considered for further analysis.

When grouping the measured data by wind speed, starting from the group 1–2 m/s, both the standard deviation and the 95% fractile values became smaller with increasing wind speed (see Table 2). A similar observation can be made for the mean values of the temperature differences. For the ventilated sensors, though, the mean values were almost constant for wind speeds from 1 to 5 m/s, which was due to their continuous mechanical ventilation. At the black facade, this resulted in a negative difference between the non-ventilated and the ventilated sensor at wind speeds above 3 m/s, indicating lower temperatures for the non-ventilated sensor. However, there is no explanation for this phenomenon at the present time.

There is no group for wind speeds above 5 m/s, as there was only one datapoint available. In any case, it can be expected that temperature differences between ventilated and non-ventilated sensors become negligible at these wind speeds.

A detailed examination of temperature variations between the ventilated and non-ventilated sensors in relation to wind direction reveals a significant impact. Predominant wind directions at the research site included WNW, North, South-East, and SSE (see Figure 16a). When winds originate from the north, sensors positioned in front of the south-facing facade are affected by wind shadows, resulting in implausible results (Figure 16b): elevated wind speeds atop the test stand roof correlate with increased temperature disparities, with ventilated sensors registering higher temperatures. Conversely, when the test facade faces winds from the SE and SSE, unventilated sensor temperatures consistently exceed those of ventilated sensors (Figure 16c). The radiation error, averaging below 0.2 degrees for wind speeds below 1 m/s, escalates to 0.56 and 0.82 degrees for wind speeds up to 2 m/s in front of the black and white facade, respectively. For wind speeds up to 4 m/s, the error decreases to 0.44 degrees in front of the black facade, holding relatively steady for sensors on the white facade. In the prevailing wind direction WNW, where the wind flows nearly parallel to the test facade, the most significant temperature differences of 0.15 degrees in front of the black facade and 0.36 degrees in front of the white facade occur at low wind speeds below 2 m/s (Figure 16d). At higher wind speeds of up to 5 m/s, these differences decrease to 0.04 degrees in front of the white facade, while turning negative in front of the black facade, indicating average temperatures recorded by ventilated sensors up to 0.24 degrees higher than those recorded by unventilated sensors.

## 4. Discussion

The findings of this study underscore the significant impact of direct solar radiation on the accuracy of temperature measurements, highlighting the crucial role of sensor ventilation influenced by prevailing wind conditions and built-in mechanical ventilation in radiation shields. In the absence of wind, non-ventilated radiation shields can lead to measurement errors of up to 5 degrees due to solar radiation. The utilization of ventilated radiation shields mitigates these errors; however, in the presence of direct irradiation on hot summer days, their radiation error can also reach up to 3 degrees. Achieving precision, with an accuracy of less than one degree, appears feasible only at high wind speeds exceeding 4 m/s.

Our observed measurement errors with a non-ventilated radiation shield align with the findings of Lin et al., who reported temperature errors of 2–4 degrees on clear, windless days at midday. This deviation corresponds to the air temperature measured with a ventilated US Climate Reference Network (CRN) shield ([7], p. 1224ff). Nakamura and Mahrt also based their evaluations on a sensor with a ventilated radiation shield, noting a radiation error of more than one degree in 2.6% of the measurement data, with individual points deviating by more than 2 degrees ([9], p. 1054). It is noteworthy that the inaccuracy of the ventilated radiation shield was not considered in the assessments of both studies.

Benchmarking the accuracy of temperature sensors against mechanically aspirated, shaded, multi-walled tubes, typically assumed to have negligible error (<0.1 °C) ([8], p. 185), is a common approach. In this study, the reference sensor under the test stand met these criteria. The accuracy of all measurements at a distance of 30 cm in front of the facade was determined in relation to this reference sensor. Both non-ventilated and ventilated sensors exhibited significant measurement errors when air temperature was measured under direct solar radiation influence. The radiation error remained within ±2.4 and ±1.3 degrees in 95% of cases for non-ventilated and ventilated radiation shields, respectively, in front of the white facade. For the black facade, the errors were ±2.0 and ±1.5 degrees in 95% of cases. This emphasized the necessity for an additional reference measurement with a ventilated sensor in a permanently shaded position when measuring temperatures in the sun, allowing subsequent data correction.

Local wind speed emerged as a crucial parameter influencing temperature measurement accuracy. According to Kurzeja [8], reliable temperature readings are attainable at a wind speed of 1 m/s. Consistent with Nakamura and Mahrt ([9], p. 1056), a radiation error of 0.6 degrees occurred at wind speeds below 1 m/s, decreasing to 0.3 degrees at higher wind speeds. Richardson et al. [16] recorded air temperature errors of up to 1.8 degrees during windless periods. In both studies, the temperature reference is a temperature sensor in a mechanically aspirated shield which itself has an accuracy of 0.3 degrees and a radiation error of 0.2 degrees at 1000 W/m^2^ [17]. In this study, the temperature differences of non-ventilated sensors compared to ventilated sensors were below 0.6 and 1.1 °C for black and white facades, respectively, for 95% of cases. The most significant temperature differences (with 95% fractiles of 1.1 and 1.6 °C, respectively) occurred at wind speeds below 2 m/s, with differences within measurement accuracy at wind speeds exceeding 4 m/s.

The proximity of sensors to the facade may impact measurement accuracy, with lower wind speeds and solar reflections potentially introducing additional radiation errors. Conversely, depending on facade orientation, sensors in close proximity may experience more shading from the building, reducing exposure to direct solar radiation and thus lowering average radiation errors. Future studies should investigate the optimal distance between sensors and facades to minimize radiation errors. Additionally, the influence of wind direction on radiation errors near facades using ventilated radiation shields should be explored, potentially employing a weather station to measure wind speed and direction near temperature sensors positioned in front of the facade.

In conclusion, this study underscores the critical importance of ventilated radiation shields for achieving high-quality temperature measurements in the sun. Nevertheless, ventilated shields can still lead to considerable radiation errors when fixed in front of a south-oriented facade. White facades may exacerbate errors due to additional reflections of solar radiation, while black facades introduce heating of the air layer, contributing to measurement errors. Future research should delve into evaluating radiation errors of different ventilated radiation shields and establishing requirements for reference temperature measurements when assessing temperatures in the sun. Establishing these measurement standards is essential for providing accurate temperature data to evaluate the impact of facade greening and other Urban Heat Island mitigation measures, crucial for guiding cities towards a sustainable future.

## Figures and Tables

**Figure 1 sensors-24-01564-f001:**
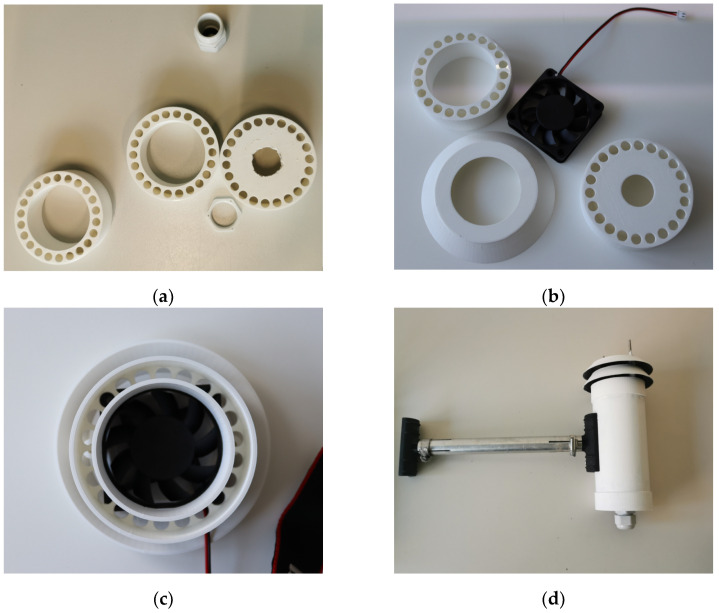
Assembly of the investigated ventilated radiation shields: (**a**) individual parts from the 3D printer; (**b**) individual parts including the fan for the lid; (**c**) assembled lid; (**d**) assembled radiation shield.

**Figure 2 sensors-24-01564-f002:**
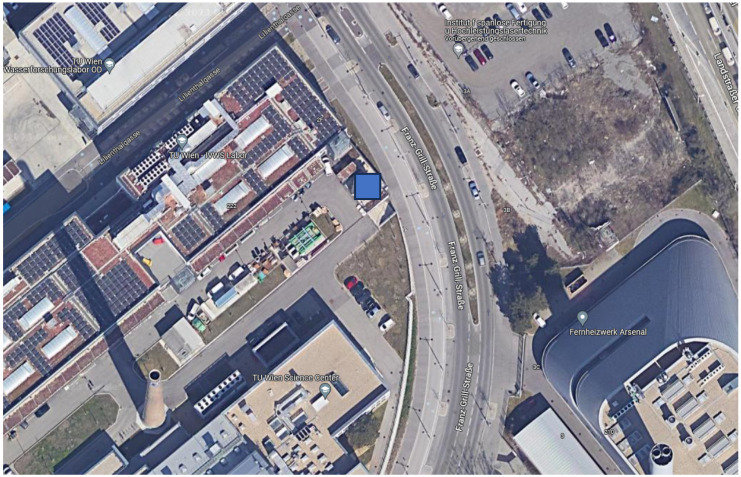
Location (blue square) and surroundings of the test stand at the Campus Science Center of the TU Wien [14].

**Figure 3 sensors-24-01564-f003:**
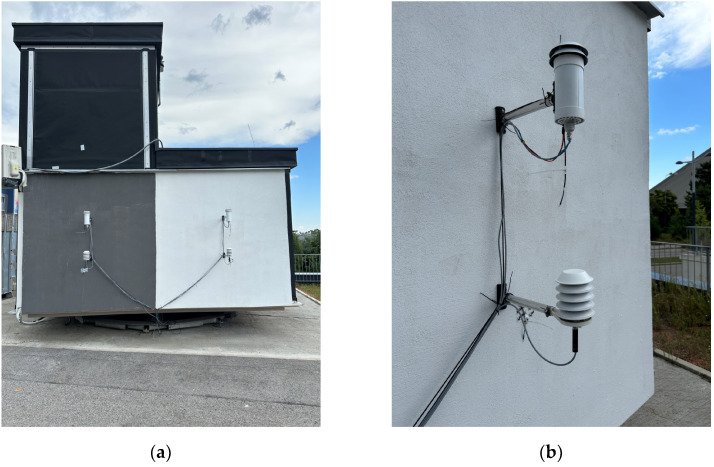
Set-up of the sensors at the test stand: (**a**) south facing facade of the test stand; (**b**) sensors on the white-painted part of the facade.

**Figure 4 sensors-24-01564-f004:**
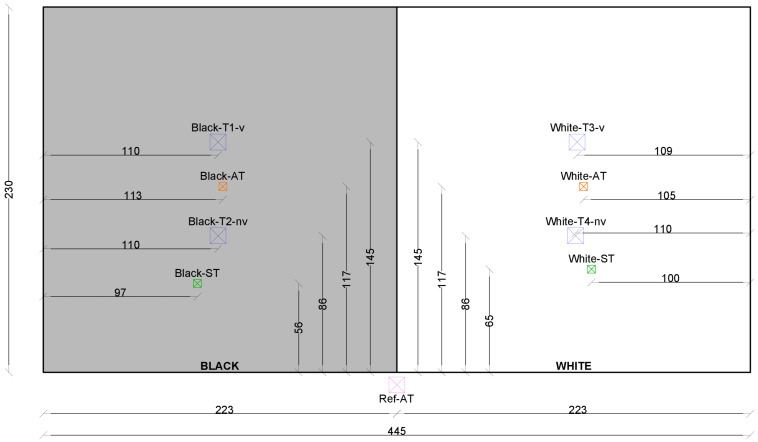
Scheme of the position of the sensors, with T1–T4 being the sensors within the ventilated (v) and non-ventilated (nv) radiation shields, AT being the air temperature sensors without a radiation shield, ST being the surface temperature sensors, and Ref-AT being the reference air temperature sensor.

**Figure 5 sensors-24-01564-f005:**
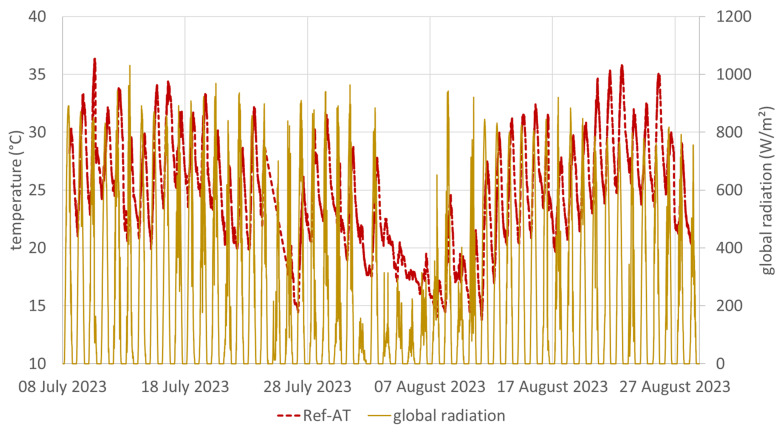
Air temperature (Ref-AT) at the test stand and global radiation of the weather station Wien Unterlaa from 8 July 2023 to 27 August 2023.

**Figure 6 sensors-24-01564-f006:**
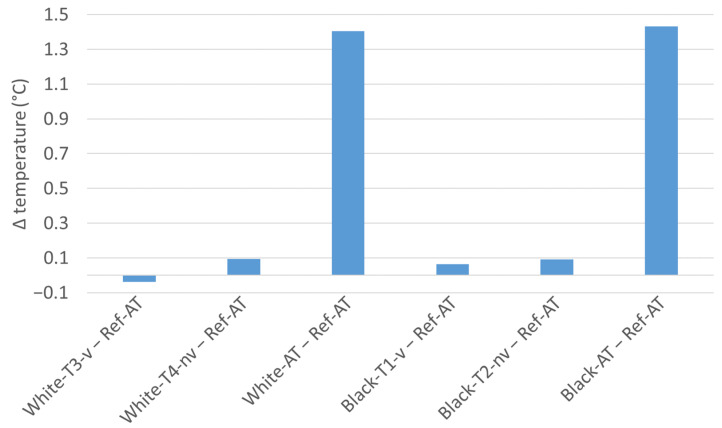
Differences in the mean temperatures of the sensors in front of the facade and the reference sensor below the test stand from 8 July 2023 to 28 August 2023.

**Figure 7 sensors-24-01564-f007:**
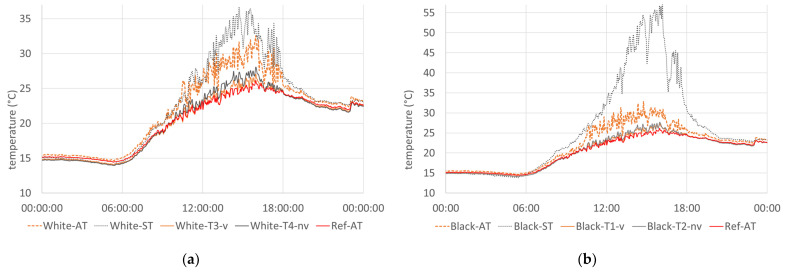
Measured temperatures on 27 July 2023: (**a**) temperatures of the sensors in front of the white facade; (**b**) temperatures of the sensors in front of the black facade.

**Figure 8 sensors-24-01564-f008:**
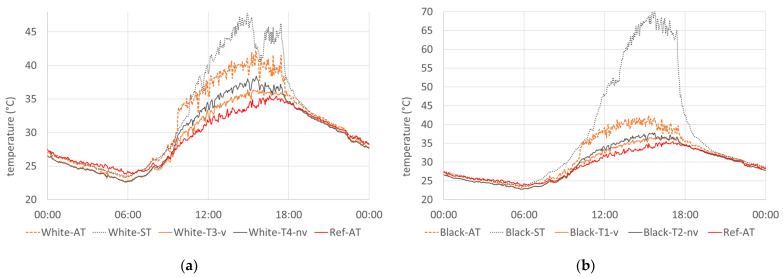
Measured temperatures on 21 August 2023: (**a**) temperatures of the sensors in front of the white facade; (**b**) temperatures of the sensors in front of the black facade.

**Figure 9 sensors-24-01564-f009:**
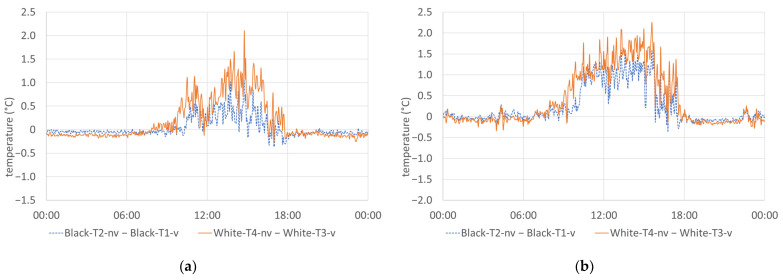
Measured temperature differences between non-ventilated and ventilated radiation shields in front of the black and the white facade: (**a**) 27 July 2023; (**b**) 21 August 2023.

**Figure 10 sensors-24-01564-f010:**
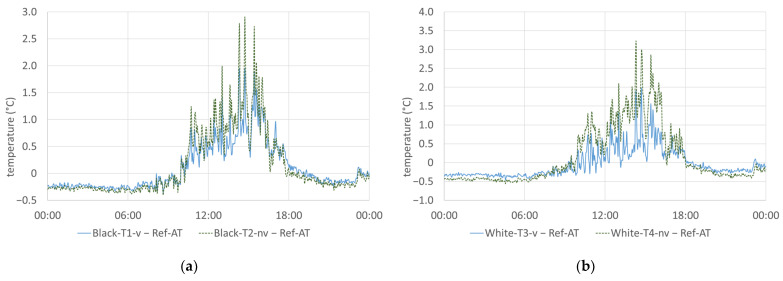
Deviation of the ventilated and non-ventilated sensors from the reference: (**a**) black facade—27 July 2023; (**b**) white facade—27 July 2023; (**c**) black facade—21 August 2023; (**d**) white facade—21 August 2023.

**Figure 11 sensors-24-01564-f011:**
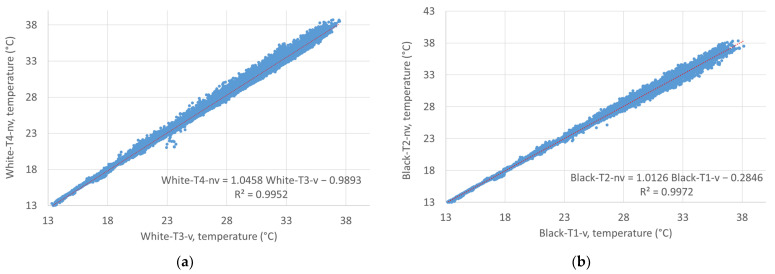
Correlation of the measurement values of the non-ventilated and ventilated sensors for the entire measurement period: (**a**) white facade; (**b**) black facade.

**Figure 12 sensors-24-01564-f012:**
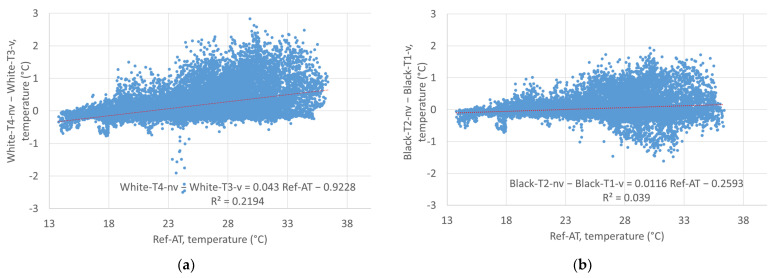
Temperature difference of the non-ventilated and ventilated sensors as a function of the reference temperature for the entire measurement period: (**a**) white facade; (**b**) black facade.

**Figure 13 sensors-24-01564-f013:**
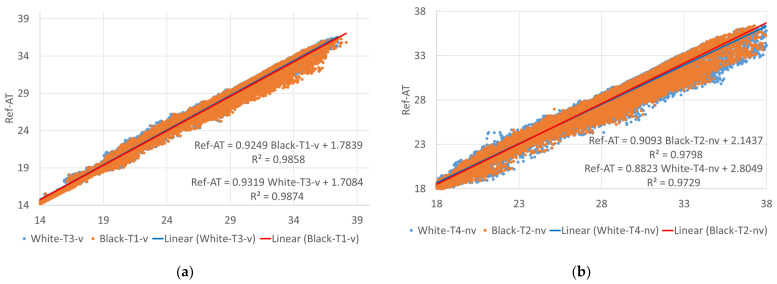
Correlation of the measurement values of the different sensors in front of the facade with the reference sensor Ref-AT below the test stand for the entire measurement period: (**a**) ventilated radiation shield; (**b**) non-ventilated radiation shield.

**Figure 14 sensors-24-01564-f014:**
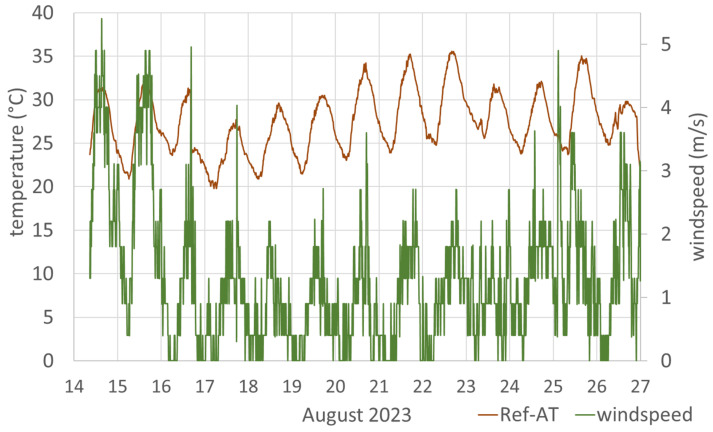
Air temperature (Ref-AT) and wind speed at the test stand from 14 August 2023 to 27 August 2023.

**Figure 15 sensors-24-01564-f015:**
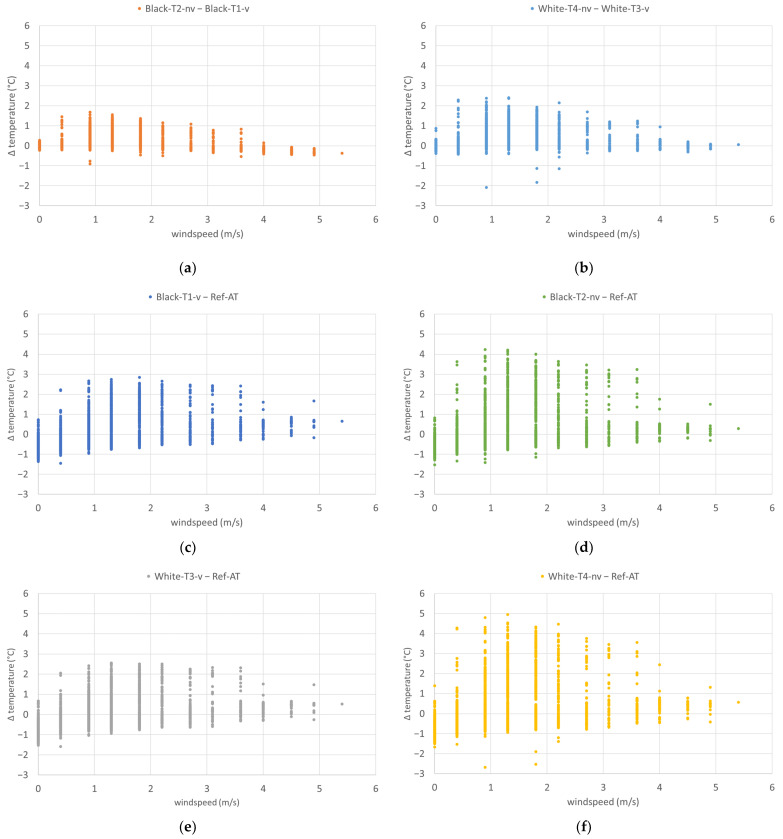
Temperature differences at different wind speeds from 14 August 2023 to 27 August 2023: (**a**) Black-T2-nv–Black-T1-v; (**b**) White-T4-nv–White-T3-v; (**c**) Black-T1-v–Ref-AT; (**d**) Black-T2-nv–Ref-AT; (**e**) White-T3-v–Ref-AT; (**f**) White-T4-nv–Ref-AT.

**Figure 16 sensors-24-01564-f016:**
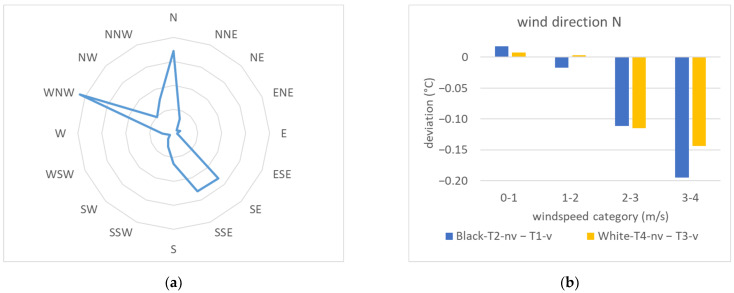
Temperature differences by wind direction with measurement data from 14 August 2023 to 27 August 2023: (**a**) wind rose; (**b**) temperature differences for north wind; (**c**) temperature differences for SE and SSE wind; (**d**) temperature differences for WNW wind.

**Table 1 sensors-24-01564-t001:** Means, standard deviations, and 95% fractiles for the relative temperature differences of the ventilated and non-ventilated sensors in front of the white and black facade, and the reference from 8 July 2023 to 28 August 2023.

Values in °C	Black-T2-nv–Black-T1-v	Black-T1-v–Ref-AT	Black-T2-nv–Ref-AT
Mean	0.03	0.07	0.09
Standard deviation	0.28	0.69	0.83
95% fractile	0.63	1.51	1.95
	**White-T4-nv–White-T3-v**	**White-T3-v–Ref-AT**	**White-T4-nv–Ref-AT**
Mean	0.13	−0.04	0.10
Standard deviation	0.44	0.64	1.01
95% fractile	1.13	1.34	2.38

**Table 2 sensors-24-01564-t002:** Means, standard deviations, and 95% fractiles of temperature differences depending on wind speed from 14 August 2023 to 27 August 2023.

Wind Speed	ms^−1^	0–1	1–2	2–3	3–4	4–5
Black-T2-nv–Black-T1-v	°C	0.09; ±0.26;	0.29; ±0.42;	0.12; ±0.36;	−0.06; ±0.26;	−0.22; ±0.12;
		0.71	1.12	0.93	0.62	−0.03
White-T4-nv–White-T3-v	°C	0.11; ±0.42;	0.49; ±0.64;	0.26; ±0.53;	0.15; ±0.36;	0.04; ±0.17;
		1.08	1.63	1.36	1.10	0.26
Black-T1-v–Ref-AT	°C	−0.23; ±0.56;	0.50; ±0.86;	0.48; ±0.88;	0.53; ±0.70;	0.49; ±0.33;
		0.92	2.10	2.23	2.21	0.85
Black-T2-nv–Ref-AT	°C	−0.15; ±0.77;	0.79; ± 1.26;	0.61; ±1.20;	0.47; ±0.91;	0.27; ±0.33;
		1.70	3.19	3.14	2.81	0.53
White-T3-v–Ref-AT	°C	−0.31; ±0.57;	0.41; ±0.86;	0.40; ±0.85;	0.42; ±0.66;	0.36; ±0.30;
		0.90	2.02	2.13	2.04	0.66
White-T4-nv–Ref-AT	°C	−0.20; ±0.92;	0.91; ±1.46;	0.65; ±1.36;	0.57; ±1.00;	0.40; ±0.41;
		1.89	3.56	3.54	3.12	0.79

## Data Availability

The raw data supporting the conclusions of this article will be made available by the authors on request. The data are not publicly available due to confidentiality reasons.

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
