# Peer review of "Measurement Errors When Measuring Temperature in the Sun"

_sensors, 2024, doi:10.3390/s24051564_

Round 1

Reviewer 1 Report

Comments and Suggestions for Authors

This is an interesting manuscript that explores the effect of ventilation on the measurement accuracy of air temperature sensors. It is easy to follow the authors’ ideas throughout the manuscript. Here are a few comments for consideration:

       1)            Please introduce Figure 2 in the main text before displaying the figure.

     2)            Please avoid commercialising by mentioning the sensors’ names. Instead, describe the sensors and mention their accuracies.

     3)            Kindly consider changing the colour code of figures 7.b and 8.b, especially for Black-ST and Black-T2-nv to enhance the figures' readability.

     4) Were the differences in the measured temperatures statistically significant? Were they attributed to random variation? Please consider performing a statistical test to explore the statistical significance of the variations.

     5)            In the correlations equations, please replace x and y with the corresponding temperatures.

     6)            Discussion: [The findings of this study underscore the significant impact of direct solar radiation on the precision of temperature measurements, …] Please replace (precision) with (accuracy). In meteorology, precision and accuracy are different.

      7)            What is the speed of the fan in the ventilated shield? Why select this speed? What is the relation between this speed and the prevailing wind speed in the measured place? Please discuss this in the manuscript.

     8)            What is the horizontal distance between the ventilated and non-ventilated shields and the facades? What can be the influence of this distance? Please discuss this in the manuscript. 

Comments on the Quality of English Language

The paper is easy to read. I detected one or two typos that needs correction. 

Reviewer 2 Report

Comments and Suggestions for Authors

This manuscript is very well written and interesting. However, there are a few mostly formal issues that should be addressed before the publication:

- please correct all references in the text to the required format. For example, "Avraham et al. (2022) [4]" should be: "Avraham et al. [4]"

- Fig. 12,13, Table 1, Table 2 please replace a decimal comma with a decimal point

- L101 - please provide information if the direction of wind was also measured. If so, add this information also to the results section as this could have an important impact on the results (the windward and leeward sides of the wall)

- please provide an accurate information on distance between the sensors and facade (or the length of the metallic stand shown in Fig. 1d)

- L135 - please provide information how far is the Unterlaa weather station ("close-by" is not enough)

-L285 typo error "cero"

-L320 please check if "p. 1224ff." is correct
